# Electrophysiologic Patterns of Symptomatic Vincristine-Induced Peripheral Neuropathy in Children with Acute Lymphocytic Leukemia

**DOI:** 10.3390/jcm12020686

**Published:** 2023-01-15

**Authors:** Jae-Gyeong Jeong, Chang-Hwan Ahn, Yu-Sun Min, Sung Eun Kim, Ji Yoon Kim, Tae-Du Jung

**Affiliations:** 1Department of Rehabilitation Medicine, Kyungpook National University Hospital, Daegu 41944, Republic of Korea; 2Department of Rehabilitation Medicine, Kyungpook National University Chilgok Hospital, Daegu 41404, Republic of Korea; 3Department of Rehabilitation Medicine, School of Medicine, Kyungpook National University, Daegu 41944, Republic of Korea; 4Department of Pediatrics, Kyungpook National University Chilgok Hospital, Daegu 41404, Republic of Korea; 5Department of Pediatrics, School of Medicine, Kyungpook National University, Daegu 41944, Republic of Korea

**Keywords:** acute lymphocytic leukemia, vincristine-induced peripheral neuropathy, chemotherapy-induced peripheral neuropathy, nerve conduction study, pediatrics

## Abstract

Acute lymphocytic leukemia is one of the most common cancers in children. Multi-drug chemotherapy is used for treatment, and the representative drug is vincristine. Although various side effects may occur due to vincristine, the association with peripheral neuropathy is high compared to that of other drugs. This study focused on children under the age of 18 years of age with ALL who received chemotherapy containing vincristine. We retrospectively analyzed the results of a nerve conduction study and a cumulative dose of vincristine in 30 children diagnosed with peripheral neuropathy. The average cumulative dose until diagnosis of vincristine-induced peripheral neuropathy was 14.99 ± 1.21 mg/m^2^, and motor nerves were predominantly involved. Additionally, a marked decrease in average amplitude was also observed in motor nerves. In addition, when the relationship between the incidence of peripheral neuropathy and the cumulative dose was analyzed through the survival curve, about 50% of children developed peripheral neuropathy at a dose of 15.5 ± 1.77 mg/m^2^. Based on the electrophysiological characteristics of pediatric vincristine-induced peripheral neuropathy, as well as the relationship between the incidence rate and the cumulative dose, it is possible to observe more closely the vincristine-induced peripheral neuropathy occurrence in children with ALL at an appropriate time.

## 1. Introduction

Acute lymphocytic leukemia (ALL) is caused by a malignant change in and the abnormal proliferation of lymphoid progenitor cells [1]. In the United States, ALL is the most common cancer in children and the second most common acute leukemia in adults [1,2]. Multi-agent chemotherapy regimens are applied for the treatment of ALL [3]. Due to the toxic effects of drugs used in chemotherapy [2], adverse effects, such as osteonecrosis, metabolic syndrome, obesity, cardiovascular impairment, and abnormalities in the central and peripheral nervous system occur, with chemotherapy-induced peripheral neuropathy being one of the most common side effects [4]. Vincristine, an essential component of the chemotherapy protocol for treating ALL in children, is a representative drug with neurotoxicity [5].

Vincristine is a plant-derived vinca alkaloid anti-tumor agent introduced in the 1960s to treat leukemia, lymphoma, and cancer [6]. It is a cell-cycle-specific anti-tumor drug that stops cell division by inhibiting the synthesis of spindle microtubules during cell division [7], causing tumor cell necrosis. As mentioned earlier, the main side effect of vincristine is neurotoxicity, which affects axonal transport, causing Wallerian degeneration of the neurons and, consequently, axonal neuropathy [8,9]. Peripheral neuropathy caused by vincristine is commonly referred to as vincristine-induced peripheral neuropathy (VIPN) [8]. VIPN can cause symptoms of motor, sensory, and autonomic neuropathy [10]. Therefore, the quality of life of patients may be reduced by dysautonomia, upper or lower extremity weakness, and sensory symptoms, such as numbness, tingling sensation, and neuropathic pain [11]. These side effects of the drug become an essential factor in adjusting the dosage of the drug [6], which may decrease or delay the therapeutic effect [8].

Although there have been several studies on VIPN, few studies have been conducted on children based on electrophysiological data. Therefore, in this study, the characteristics of VIPN in children with ALL were quantitatively analyzed based on the electrophysiological pattern of VIPN using nerve conduction study (NCS) results and the cumulative dose of vincristine that had been administered until the diagnosis of VIPN.

## 2. Materials and Methods

### 2.1. Subjects

This study included patients who visited Chilgok Kyungpook National University Hospital between January 2012 and December 2021, were diagnosed with ALL, and received chemotherapy. The department of Rehabilitation Medicine at our center performed an NCS on children with ALL who were consulted from the pediatric department when patients displayed symptoms of peripheral neuropathy after chemotherapy containing vincristine. Therefore, we inevitably retrospectively investigated the information of only symptomatic patients with abnormal findings in the NCS. The data included the chemotherapy protocol used, cumulative dose of vincristine until the NCS, and results of the NCS. In total, 30 patients were analyzed based on the inclusion criteria. The inclusion criteria were as follows: (1) patients who were younger than 18 years of age; (2) patients diagnosed with only ALL, and not other types of leukemia; and (3) patients who had one or more nerve abnormalities in the NCS. The exclusion criteria were as follows: (1) patients who had peripheral polyneuropathy before chemotherapy; (2) patients who had underlying diseases that could influence the NCS results, such as diabetes; and (3) patients who had unstable vital signs.

### 2.2. ALL Treatment Protocol

In this study, the pediatric patients with ALL were treated according to the Children’s Oncology Group (COG) guidelines, while twenty patients were treated according the AALL0331 protocol, seven patients were treated according to the AALL0232 protocol, two patients were treated according to the AALL0434 protocol, and one patient was treated according the CCG 1961 protocol. Each process is described below.

#### 2.2.1. AALL0331 Protocol

In the induction process, patients received an intrathecal (IT) cytarabine on day 1; weekly intravenous (IV) vincristine (VCR) for 4 doses; oral dexamethasone for 28 days; intramuscular PEG on day 4, 5, or 6; and IT methotrexate (MTX) for 2 to 4 doses. In the intensified consolidation process, patients received cyclophosphamide on day 1 and 29; IV vincristine on day 15, 22, 43, and 50; cytarabine on day 1 to 4, 8 to 11, 29 to 32, and 36 to 39; PEG on day 15 and 43; mercaptopurine on day 1 to 14 and 29 to 42; and IT MTX on day 1, 8, 15, and 22. In the augmented interim maintenance process, patients received IV vincristine and IV MTX on day 1, 11, 21, 31, and 41; PEG on day 2 and 22; and IT MTX on day 1 and 31. In the augmented delayed intensification, patients received vincristine and doxorubicin on day 1, 8, and 15; PEG on day 4, 5, or 6; dexamethasone on day 1 to 7 and 15 to 21; and IT MTX on day 1, 29, and 36. The following process was the augmented interim maintenance II, which was the same as augmented interim maintenance I. The final process was augmented delayed intensification II, which was the same as augmented delayed intensification I.

#### 2.2.2. AALL0232 Protocol

In the induction process, patients received IT cytarabine on day 1; weekly IV vincristine and daunorubicin; oral or IV prednisone for 28 days; and IM PEG on day 4, 5, or 6. In the consolidation process, patients received IV cyclophosphamide on day 1 and 29; IV cytarabine on day 1 to 4, 8 to 11, 29 to 32, and 36 to 39; mercaptopurine on day 1 to 14 and day 29 to 42; IV vincristine on day 15, 22, 43, and 50; PEG on day 15 and 43; and IT MTX on day 1, 8, 15, and 22. In the interim maintenance 1 process, patients received IV vincristine and IV MTX on day 1, 15, 29, and 43; oral mercaptopurine for 56 days; and IT MTX on day 1 and 29. In the delayed intensification 1 process, patients received IV vincristine on day 1, 8, 15, 43, and 50; PEG on day 4, 5 or 6 and day 43; dexamethasone on day 1 to 7 and 15 to 21; doxorubicin on day 1, 8, and 15; cytarabine on day 29 to 32 and day 36 to 39; cyclophosphamide on day 29; thioguanine on day 29 to 42; and IT MTX on day 1, 29, and 36. In the interim maintenance II process, patients received IV vincristine and IV MTX on day 1, 11, 21, 31, and 41; PEG on day 2 and 22; and IT MTX on day 1 and 31. Furthermore, the delayed intensification II process was the same as the delayed intensification I process. In the maintenance therapy, patients received vincristine on day 1, 29, and 57; prednisone on day 1 to 5, 29 to 33, and 57 to 61; oral mercaptopurine daily; oral MTX weekly; and IT MTX on day 1 and 29. Maintenance therapy consisted of repeated 84-day cycles. The total duration of therapy was 2 years for female patients and 3 years for male patients from the start of interim maintenance I.

#### 2.2.3. AALL0434 Protocol

In the induction process, patients received IT cytarabine on day 1; weekly IV vincristine; IV daunorubicin for 4 weeks; IV prednisone for 28 days; PEG on day 4, 5, or 6; and IT MTX on day 8 and 29. In the consolidation process, patients received cyclophosphamide on day 1 and 29; cytarabine on day 1 to 11 and 29 to 39; mercaptopurine on day 1 to 14 and 29 to 42; vincristine on day 15, 22, 43, and 50; and PEG on day 15 and 43. In the interim maintenance process, patients received IV vincristine and MTX on day 1, 11, 21, 31, and 41; PEG on day 2 and 22; and IT MTX on day 1 and 31. In the delayed intensification process, patients received vincristine on day 1, 8, 15, 43, and 50; PEG on day 4, 5, or 6 and 43; dexamethasone on day 1 to 7 and 15 to 21; doxorubicin on day 1, 8, and 15; cytarabine on day 29 to 32 and 36 to 39; cyclophosphamide on day 29; thioguanine on day 29 to 42; and IT MTX on day 1, 29, and 36. In the maintenance process, patients received IV vincristine on day 1, 29, and 57; prednisone on day 1 to 5, 29 to 33, and 57 to 61; mercaptopurine for 84 days; oral MTX every week; and IT MTX on day 1 and 29. Maintenance consisted of repeated 84-day cycles. The total duration of therapy was 2 years for female patients and 3 years for male patients from the start of interim maintenance.

#### 2.2.4. CCG1961 Protocol

In the induction process, patients received weekly IV vincristine and IV daunorubicin for 4 weeks; prednisone for 28 days; 6 doses of L-Asparaginase every other day from day 9 to 22; IT cytarabine on day 1; and IT MTX on day 8 and 29. In the consolidation process, patients received cyclophosphamide on day 1 and 15; IV cytarabine on day 1 to 4, 8 to 11, 15 to 18, and 22 to 25; oral mercaptopurine for 28 days; and IT MTX on day 1, 8, 15, and 22. In the interim maintenance process, patients received oral mercaptopurine for 42 days; oral MTX on day 7, 14, 21, 28, and 35; and IT MTX on day 1 and 29. In the delayed intensification process, patients received oral dexamethasone on day 1 to 21; IV vincristine and doxorubicin on day 1, 8, and 15; 6 doses of L-asparaginase every other day from day 4; IT MTX on day 1, 29, and 36; cyclophosphamide on day 29; thioguanine on day 29 to 42; and IV cytarabine on day 29 to 32 and 36 to 39. In the maintenance process, patients received IV vincristine on day 1, 29, and 57; oral prednisone on day 1 to 5, 29 to 33, and 57 to 61; mercaptopurine for 84 days; oral MTX weekly for 12 weeks; and IT MTX on day 1 and 29. Maintenance consisted of repeated 84-day cycles. The total duration of therapy was 2 years for female patients and 3 years for male patients from the start of interim maintenance I.

### 2.3. Electrophysiological Studies

The NCS was performed using a *Synergy* (Oxford Medelec, Wiesbaden, Germany) machine. For the NCS, the median and ulnar nerves in the upper extremities and the peroneal and tibial nerves in the lower extremities were examined for each section as motor nerves. As sensory nerves, the median and ulnar nerves in the upper extremities and the superficial peroneal and sural nerves in the lower extremities were examined. In motor nerves, the amplitude and distal latency of the compound muscle action potential (CMAP) and conduction velocity between stimulation points were measured. In sensory nerves, the peak latency and amplitude of the sensory nerve action potential (SNAP) were measured. Distal latency was measured in milliseconds as the time from the stimulation of the motor nerve to the onset of CMAP, and conduction velocity was measured in meters per second. Amplitude was measured from the negative peak to the positive peak, expressed in millivolts for motor nerves and microvolts for sensory nerves.

For the interpretation of the NCS, the normal reference values for the pediatric age group suggested by Cai and Zhang [12] were applied. For quantitative analysis, the degree of change in the CMAP and SNAP amplitude compared to normal reference values was expressed as a percentile (change in amplitude [%] = measured value × 100/reference value). In addition, the types of nerves affected in each patient and the number of nerves affected among the four motor and four sensory nerves tested were analyzed, respectively.

### 2.4. Statistical Analyses

For statistical analyses, *SPSS version 23.0* (IBM SPSS Statistics, Version 23.0., IBM Corp., Armonk, NY, USA) was used. The chi-squared test was used to analyze the difference in the number of affected motor and sensory nerves. In addition, Student’s *t*-test and the nonparametric Mann–Whitney U test were used to analyze the degree of change (%) in the CMAP and SNAP amplitudes for each affected nerve in the motor and sensory nerves. The significance level was set at *p* < 0.05. In addition, the Kaplan–Meier survival analysis method was used to assess the correlation between the cumulative dose of vincristine and the incidence of VIPN. The proportion of patients who developed VIPN was analyzed in relation to the cumulative vincristine dose.

## 3. Results

### 3.1. Demographics

The demographic data of the patients are presented in Table 1. There were 15 boys and 15 girls, and the average age was 7.63 ± 4.69 years. The average cumulative dose of vincristine, until abnormal findings were detected in the NCS, was 14.99 ± 1.21 mg/m^2^. The average period from the first administration of vincristine to the discovery of abnormal findings in the NCS was 143.37 ± 74.02 days.

### 3.2. Electrophysiological Findings

When the NCS was performed in all patients, abnormal findings of one or more motor or sensory nerves were found. All patients had abnormalities in two or more motor nerves, and 14 patients had abnormal findings in one or more sensory nerves. The proportion of the affected nerves among the motor and sensory nerves was 85.83% ± 18.20% and 22.5% ± 29.62%, respectively, showing a statistically significant difference (*p* < 0.001).

Abnormalities were found in motor nerves in 20 patients and in sensory nerves in 5 patients in cases involving the median nerve. In cases where the ulnar nerve was involved, 25 patients exhibited motor nerve abnormalities, and 4 patients showed sensory nerve abnormalities. In cases including the peroneal nerve, 30 patients exhibited motor nerve abnormalities, and 5 patients exhibited sensory nerve abnormalities. In cases involving the tibial nerve (motor), 28 patients had motor nerve abnormalities, whereas in cases involving the sural nerve (sensory), 13 patients had sensory nerve abnormalities. A significantly higher frequency of motor nerve abnormalities was observed in all types of nerves (*p* < 0.001), and motor nerve abnormalities were observed in all patients in the case of peroneal nerve involvement (Table 2).

The average change in CMAP amplitude compared to the normal reference value was 87.63% ± 41.66% in the median nerve, 74.89% ± 34.16% in the ulnar nerve, 10.8% ± 11.16% in the peroneal nerve, and 65.5% ± 25.54% in the tibial nerve. All types of nerves demonstrated a decrease in CMAP amplitude. However, the average change in SNAP amplitude did not show a decreasing pattern. The average change in SNAP amplitude compared to the normal reference value was 185.68% ± 89.29% in the median nerve, 220.02% ± 97.64% in the ulnar nerve, 241.27% ± 118.77% in the superficial peroneal nerve, and 122.30% ± 47.16% in the sural nerve. A significant difference was confirmed when the rate of change in the CMAP amplitude of all motor nerves and the rate of change in the SNAP amplitude of all sensory nerves were compared. (*p* = 0.000; Table 3).

When the motor NCS results were analyzed qualitatively, a length-dependent pattern could be confirmed in all patients. In addition, in the motor NCS, axonal-type neuropathy was observed rather than demyelinating-type neuropathy through findings that were not accompanied by the prolongation of the distal latency, temporal dispersion, and conduction block.

### 3.3. Cumulative Dose of Vincristine up to Diagnosis of Symptomatic VIPN

We used the Kaplan–Meier method to assess the cumulative dose of vincristine administered until the onset of VIPN (Figure 1). According to the aforementioned statistical analysis, symptomatic VIPN occurred in approximately 50% of the patients when the cumulative dose of vincristine administered was 15.5 ± 1.77 mg/m^2^. When analyzed using the survival table, symptomatic VIPN occurred in approximately 25% and 75% of the patients at a cumulative vincristine dose of 10.48 ± 2.42 and 19.21 ± 2.95 mg/m^2^, respectively. 

## 4. Discussion

Vincristine is one of the most important anti-tumor agents used to treat pediatric ALL [2]. It binds to tubulin, which forms microtubules in cells and stops the division of cancer cells during metaphase [13]. One of its most common side effects is neurotoxicity [4]. Although the mechanism of neurotoxicity caused by vincristine has not been fully elucidated, it is thought that the binding of vincristine interferes with axonal transport and the secretory activity of neurons, causing axonal degeneration [14,15]. Previous studies using rats have shown pathological changes, such as the degeneration of some motor neurons in the form of axonal shrinkage [16]. To continue treatment, the patient must be protected from these side effects. Although VIPN can be clinically inferred from the neurological symptoms in children receiving vincristine, it is important to objectively diagnose the patient’s condition through an NCS and establish an appropriate management plan.

This study quantitatively analyzed the characteristics of VIPN in Korean children aged under 18 years who received vincristine chemotherapy for ALL. When analyzing the results of the NCS performed on a child suspected to have peripheral neuropathy, the rate of motor nerve involvement was significantly higher than that of sensory nerve involvement. In addition, considering that the average amplitude decrease rate was higher in motor nerves, it could be inferred that the motor nerves were damaged more severely. Abnormal findings were observed in all patients with peroneal motor nerve involvement. All patients showed a length-dependent neuropathy pattern. There were no abnormal findings, such as the prolongation of distal latency, reduction in conduction velocity, and temporal dispersion, appropriate for axonal-type neuropathy.

In general, peripheral neuropathy occurring after vincristine administration mainly involves sensory nerves [17,18], but in this study, motor nerves were more often involved. This is a characteristic of pediatric VIPN that differs from the characteristics of adult VIPN, and we can consider the temporal difference in myelination in the maturation process of the central and peripheral nerves in the developmental process [19]. It is known that the maturation and development of sensory nerves precede motor nerves [20,21]. Therefore, it is presumed that motor nerves, which are in a relatively immature myelination process, are susceptible to damage by drugs, such as vincristine, that cause axon damage, which explains the remarkable findings of characteristic motor nerve invasion in children [19].

Previous studies have reported that the characteristics are more similar to axonal-type neuropathy than to demyelinating-type neuropathy [8,22,23], which was confirmed in the NCS of children with VIPN in this study. Given that the electrophysiological characteristics are similar to acute motor axonal neuropathy, a variant of Guillain–Barre syndrome, the effect of vincristine on nerves in children may be similarly related to the mechanisms of channelopathy.

Additionally, this study analyzed the cumulative incidence of VIPN according to the cumulative dose of vincristine using the survival curve. According to the analysis, peripheral neuropathy occurred in 50% of the patients when 15.5 ± 1.77 mg/m^2^ of vincristine was administered cumulatively. Several previous studies have also found that the incidence of VIPN increases in proportion to the dose of vincristine used in children diagnosed with VIPN [24,25,26]. However, this study presented a specific value of VIPN incidence according to the cumulative dose of vincristine for which the survival curve was applied. Hence, the objective nature of the analysis might be considered an advantage in the present study compared to previous studies.

Chemotherapy-induced peripheral neuropathy (CIPN) is distinct from cancer pain when cancer directly invades nerve tissue and compresses nerves or forms bone metastasis. It is observed in about 40% of patients receiving chemotherapy [17]. In general, if peripheral neuropathy develops after chemotherapy, the symptoms will disappear within a few months after treatment ends. However, in rare cases symptoms may last longer [27]. Treatment for peripheral neuropathy is based on symptoms. If sensory symptoms such as neuropathic pain or paresthesia are the main symptoms, medications, such as gabapentin, tricyclic antidepressants, pyridoxine, pyridostigmine, and glutamic acid, may be used [28]. In addition, if motor symptoms such as weakness are the main symptoms, physical therapy interventions (strength, endurance, and balance training) can be performed [29].

The significance of the results of this study is as follows. First, in general, it is easy to observe abnormalities in Aα and Aβ fibers, which are large-diameter myelinated motor nerve fibers, when performing NCS. However, the disadvantage is that it does not reflect the lesions of small nerve fibers, such as Aδ and C fibers [30]. In children, because movement disorders and the involvement of motor nerves appear relatively early and severely, NCS can be useful because NCS preferentially represents motor nerve lesions, which are large nerve fibers. Further, this study analyzed the cumulative incidence of VIPN according to the cumulative dose of vincristine and the electrophysiological characteristics of pediatric VIPN. Based on these data, it is more meaningful to closely monitor the patient’s condition when vincristine is administered in the cumulative dose range, in which the risk of VIPN is increased, and to suggest an appropriate period to conduct an NCS.

There are several limitations in this study. First, the pediatric patients with ALL in this study were treated with the AALL0331 protocol in twenty patients, the AALL0232 protocol in seven patients, the AALL0434 protocol in two patients, and the CCG1961 protocol in one patient. These are the protocols for multi-drug combination chemotherapy. In addition to vincristine, drugs, such as dexamethasone, 6-mercaptopurine, methotrexate, doxorubicin, and asparaginase, were also used concomitantly. Therefore, it is difficult to conclude that peripheral neuropathy as a side effect was caused only by vincristine. However, looking at the typical side effects according to the mechanism of action of each drug, confined to the neuromuscular system, dexamethasone can cause myopathy, methotrexate is toxic to the central nervous system, and doxorubicin is known to cause cardiomyopathy. It is universally known that the main side effect of vincristine is peripheral neuropathy, even within the appropriate therapeutic dose [17]. Many drugs can cause peripheral neuropathy, including vinorelbine, vinblastine, carboplatin, cisplatin, paclitaxel, docetaxel and cabazitaxel. However, the protocols for AALL0331, AALL0232, AALL0434 and CCG1961, which were performed in the treatment of ALL in this study, did not use drugs other than these vincristines, so the effects of drugs other than vincristine were not considered. Based on this theoretical background, it was judged that vincristine had a dominant effect on the development of peripheral neuropathy compared to the other drugs used concomitantly. Therefore, in this study, the electrophysiological characteristics of chemotherapy-induced peripheral neuropathy were analyzed by limiting the effect of vincristine only. In follow-up studies, it is necessary to further analyze the results of interactions with other drugs. Second, this study retrospectively analyzed NCS data from children who showed symptoms of peripheral neuropathy while receiving chemotherapy. Asymptomatic VIPN patients were not included in our study. If they were included, the results of our study might have been different. Therefore, for an accurate analysis of VIPN incidence according to cumulative dose, analyses of data from an NCS performed regularly, regardless of neurological symptoms, are required. Therefore, it is necessary to set a certain cumulative dose or time interval and conduct the test at an appropriate time. If a prospective study that supplements these limitations is undertaken in the future, more meaningful results may be obtained. Third, it would be better if additional studies were conducted on the changes in NCS results or improvement in neurological symptoms in children whose vincristine dose was adjusted after the diagnosis of VIPN.

## 5. Conclusions

It was confirmed that VIPN that developed in children was motor-dominant axonal-type neuropathy, and the incidence increased as the administered vincristine dose increased. In addition, VIPN occurred in approximately 50% of the children included in the study at a cumulative vincristine dose of 15.5 ± 1.77 mg/m^2^. VIPN causes a delay in treatment and lowers the quality of life of children. Therefore, if VIPN is diagnosed in the early period of chemotherapy through regular NCSs and appropriate management is performed, the quality of treatment can be improved.

## Figures and Tables

**Figure 1 jcm-12-00686-f001:**
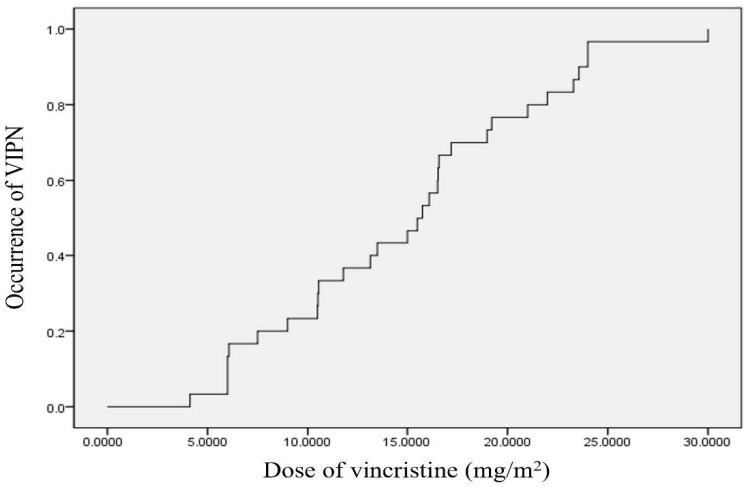
Kaplan–Meier curves for the relationship between the cumulative dose of vincristine and occurrence of symptomatic VIPN. The median dose of vincristine until onset symptoms of VIPN is 15.5 ± 1.77 mg/m^2^. VIPN, vincristine-induced peripheral neuropathy.

**Table 1 jcm-12-00686-t001:** Clinical characteristics of patients (N = 30).

	Patients
**Total (n)**	30
**Mean age (years)**	7.63 ± 4.69
**Sex**	
Male	15
Female	15
**CTx protocol**	
AALL 0331	20
AALL 0232	7
AALL 0434	2
CCG 1961	1
**Time interval between the first VCN injection and EDx examination (days)**	143.37 ± 74.02
**Cumulative dose (mg/m^2^)**	14.99 ± 1.21

CTx, chemotherapy; VCN, vincristine; EDx, electrodiagnostic.

**Table 2 jcm-12-00686-t002:** Involvement pattern of nerves in vincristine-induced peripheral neuropathy in childhood.

Motor Nerve	Number of Involved Nerves	Sensory Nerve	Number of Involved Nerves	*p*-Value
Median	20/30	Median	5/30	
Ulnar	25/30	Ulnar	4/30	
Peroneal	30/30	Superficial peroneal	5/30	
Tibial	28/30	Sural	13/30	
Total	103/120	Total	27/120	<0.001 *

* *p* < 0.05.

**Table 3 jcm-12-00686-t003:** Quantitative analysis of CMAP and SNAP amplitude in motor and sensory nerves in vincristine-induced peripheral neuropathy in pediatric acute lymphocytic leukemia.

Motor Nerve	CMAP Amplitude (%)	Sensory Nerve	SNAP Amplitude (%)	*p*-Value
Median	87.63 ± 41.66	Median	185.68 ± 89.29	
Ulnar	74.89 ± 34.16	Ulnar	220.02 ± 97.64	
Peroneal	10.8 ± 11.16	Superficial peroneal	241.27 ± 118.77	
Tibial	65.5 ± 25.54	Sural	122.30 ± 47.16	
Total	57.72 ± 42.03	Total	191.34 ± 103.24	<0.001 *

CMAP, compound motor action potential; SNAP, sensory nerve action potential; * *p* < 0.05.

## Data Availability

Available upon reasonable request.

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
