# Peer review of "Electrophysiologic Patterns of Symptomatic Vincristine-Induced Peripheral Neuropathy in Children with Acute Lymphocytic Leukemia"

_jcm, 2023, doi:10.3390/jcm12020686_

Round 1

Reviewer 1 Report

The reviewers identified an original paper examining conduction velocity testing of motor and sensory nerves to identify peripheral neuropathy in children with acute lymphocytic leukaemia requiring treatment with various anticancer drugs.
First of all, this is a study with originality, and we commend it for this assessment.
However, the reviewers cannot accept the article for publication without a clear answer to the following question. The authors should carefully review the reviewers' comments and make any necessary corrections to the manuscript.

Major1: The usual treatment of paediatric ALL should be a protocol treatment combining a number of drugs. Are the chemotherapeutic agents, immunotherapeutic agents and anticancer drugs used for ALL in the case ALL alone?

Major2: How can the side-effects of vincristine be proven if the patient is being treated with multiple anti-cancer drugs? If the article is not a study of vincristine alone, the reviewers and readers will be aware that there are questions about whether it is really a side effect of vincristine or of other drugs.

Major3: There are a number of anti-cancer drugs that cause peripheral neuropathy. In addition to vincristine, other anticancer drugs include vinorelbine, vinblastine, carboplatin, cisplatin, paclitaxel, docetaxel, cabazitaxel, vinorelbine, among others. There are also a number of molecularly targeted drugs and immunotherapeutics that can cause peripheral neuropathy. How do the authors assess these effects?

Major4: Please describe the treatment protocol for ALL involved in the case.

Minor1: Please describe any ideas the authors have to avoid peripheral neuropathy in the treatment of ALL.

Minor2: Please provide additional information on the usual prognosis for peripheral neuropathy and how it is treated.

Best regards,

Dr. Reviewer 

Author Response

Response to Reviewer 1 Comments

Dear Reviewer #1,

Thank you very much for your kind letter and comments concerning our manuscript entitled Electrophysiologic patterns of Vincristine induced Peripheral Neuropathy in Children with ALL”. First of all, thank you very much for valuing the strengths of our article.

The comments were valuable and very helpful in critically revising and improving our paper. We have discussed the comments carefully and have made corresponding corrections which we hope will be met with approval. The following are responses to your comments:

Major 1: The usual treatment of paediatric ALL should be a protocol treatment combining a number of drugs. Are the chemotherapeutic agents, immunotherapeutic agents and anticancer drugs used for ALL in the case ALL alone?

Response 1: Thank you for your valuable comments. As you said, various drugs such as Cytarabine, Dexamethasone, Vincristine, PEG, MTX, etc. are used to treat ALL. However, these drugs are also used for various cancers such as AML, CML and Non-Hodgkin’s lymphoma in addition to ALL, and there is no single drug for ALL alone.

Major 2: How can the side-effects of vincristine be proven if the patient is being treated with multiple anti-cancer drugs? If the article is not a study of vincristine alone, the reviewers and readers will be aware that there are questions about whether it is really a side effect of vincristine or of other drugs

Response 2: Thank you for your valuable comments. I agree with you that it is difficult to conclude that peripheral neuropathy as a side effect was caused only by vincristine. In response to your comments, we describe in more detail in the “Discussion section” why in this study, the electrophysiological characteristics of chemotherapy-induced peripheral neuropathy were analyzed by limiting the effect of vincristine only

(Page 8-9) Discussion, lines 343-354

There are several limitations in this study. First, the pediatric patients with ALL in this study were treated with the AALL0331 protocol in 20 patients, AALL0232 protocol in seven patients, AALL0434 protocol in two patients, and CCG1961 protocol in one patient. These are the protocols for multidrug combination chemotherapy. In addition to vincristine, drugs, such as dexamethasone, 6-mercaptopurine, methotrexate, doxorubicin, and asparaginase, were also used concomitantly. Therefore, it is difficult to conclude that peripheral neuropathy as a side effect was caused only by vincristine. However, looking at the typical side effects according to the mechanism of action of each drug, confined to the neuromuscular system, Dexamethasone can cause myopathy, Methotrexate is toxic to the central nervous system, and Doxorubicin is known to cause cardiomyopathy. It is universally known that the main side effect of vincristine is peripheral neuropathy even within the appropriate therapeutic dose [17]. Based on this theoretical background, it was judged that vincristine had a dominant effect on the development of peripheral neuropathy compared to the other drugs used concomitantly.

[17]  Argyriou, A.A.; Bruna, J.; Marmiroli, P.; Cavaletti, G. Chemotherapy-induced

peripheral neurotoxicity (CIPN): an update. Critical reviews in oncology/hematology 2012, 82,

51-77.

Major 3: There are a number of anti-cancer drugs that cause peripheral neuropathy. In addition to vincristine, other anticancer drugs include vinorelbine, vinblastine, carboplatin, cisplatin, paclitaxel, docetaxel, cabazitaxel, among others. There are also a number of molecularly targeted drugs and immunotherapeutics that can cause peripheral neuropathy. How do the authors assess these effects?

Response 3: Thank you for your valuable comments. As you say, many drugs such as vinorelbine, vinblastine, carboplatin, cisplatin, paclitaxel, docetaxel, cabazitaxel, etc. can cause peripheral neuropathy. However, in this study, in the AALL0331, AALL0232, AALL0434, and CCG 1961 protocols that we conducted for ALL treatment, these drugs other than vincristine were not used, so the effects of drugs other than vincristine were not considered.

Major 4: Please describe the treatment protocol for ALL involved in the case.

Response 4: Thank you for your valuable comments. In response to your comments, we describe the treatment protocol for ALL involved in the case in “Materials and Methods” section.

(page 2-4) Materials and Methods, lines 94-167

2.2 ALL treatment protocol

In this study, the pediatric patients with ALL were treated according to The Children’s Oncology Group(COG) guideline, and twenty patients were treated with the AALL0331 protocol, seven patients with the AALL0232 protocol, two patients with the AALL0434 protocol, and one patient with the CCG 1961 protocol. Each process is described below

2.2.1 AALL0331 protocol

In the induction process, Patients received a intrathecal(IT) cytarabine on day 1, weekly  intravenous (IV) vincristine (VCR) for 4 doses, oral dexamethasone for 28 days, intramuscular PEG on day 4, 5, or 6, and IT methotrexate (MTX) for 2 to 4 doses. In the intensified consolidation process, Patients received cyclophosphamide on day 1 and 29, IV vincristine on day 15, 22, 43 and 50, Cytarabine on day 1 to 4, 8 to 11, 29 to 32, and 36 to 39, PEG on day 15 and 43, Mercaptopurine on day 1 to 14 and 29 to 42 and IT MTX on day 1, 8, 15 and 22. In the augmented interim maintenance process, Patents received IV vincristine and IV MTX on day 1, 11, 21, 31 and 41, PEG on day 2 and 22 and IT MTX on day 1 and 31. In the augmented delayed intensification, Patients received vincristine and Doxorubicin on day 1, 8 and 15, PEG on day 4,5 or 6, Dexamethasone on day 1 to 7 and 15 to 21 and IT MTX on day 1, 29 and 36. Next process is augmented interim maintenance II same as augmented interim maintenance I. Final process is augmented delayed intensification II same as augmented delayed intensification I.

2.2.2 AALL0232 protocol

In the induction process, Patients received IT cytarabine on day 1, weekly IV vincristine and daunorubicin, Oral or IV prednisone for 28days, and IM PEG on day 4,5 or 6.  In the consolidation process, Patients received IV cyclophosphamide on day 1 and 29, IV cytarabine on day 1 to 4, 8 to 11, 29 to 32 and 36 to 39, mercaptopurine on day 1 to 14, 29 to 42, IV vincristine on day 15, 22, 43 and 50, PEG on day 15 and 43 and IT MTX on day 1, 8, 15 and 22. In Interim Maintenance 1 process, Patients received IV vincristine and IV MTX on day 1, 15, 29 and 43, Oral mercaptopurine for 56 days, and  IT MTX on day 1 and 29. In Delayed Intensification 1 process, Patients received IV vincristine on day 1, 8, 15, 43 and 50, PEG on day 4 or 5 or 6, and 43, dexamethasone on day 1 to 7 and 15 to 21, doxorubicin on day 1, 8 and 15, cytarabine on day 29 to 32 and 36 to 39, cyclophosphamide on day 29, thioguanine on day 29 to 42 and IT MTX on day 1, 29 and 36. In the Interim maintenance II process, Patients received IV vincristine and IV MTX on day 1, 11, 21, 31 and 41, PEG on day 2 and 22, IT MTX on day 1 and 31. Next the Delayed intensification II process is same as Delayed intensification I. In the Maintenance therapy, Patients received vincristine on day 1, 29 and 57, prednisone on day 1 to 5, 29 to 33 and 57 to 61, Oral mercaptopurine daily, Oral MTX weekly and IT MTX on day 1 and 29. Maintenance consists of repeated 84-day cycles, the total duration of therapy is 2 years for female patients and 3 years for male patients from the start of Interim Maintenance I.

2.2.3 AALL0434 protocol

In the induction process, Patients received IT cytarabine on day 1, weekly IV vincristine and IV daunorubicin for 4 weeks, IV prednisone for 28 days, PEG on day 4, 5 or 6 and IT MTX on day 8 and 29. In the Consolidation process, Patients received cyclophosphamide on day 1 and 29, cytarabine on day 1 to 11, 29 to 39, mercaptopurine on day 1 to 14 and 29 to 42, vincristine on day 15, 22, 43 and 50, PEG on day 15 and 43. In the Interim Maintenance process, Patients received IV vincristine and MTX on day 1, 11, 21, 31 and 41, PEG on day 2 and 22 and IT MTX on day 1 and 31. In the Delayed Intensification process, Patients received vincristine on day 1, 8, 15, 43 and 50, PEG on day 4 or 5 or 6 and 43, dexamethasone on day 1 to 7 and 15 to 21, doxorubicin on day 1, 8 and 15, cytarabine on day 29 to 32 and 36 to 39, cyclophosphamide on day 29, thioguanine on day 29 to 42 and IT MTX on day 1, 29 and 36. In the Maintenance process, Patients received IV vincristine on day 1, 29 and 57, prednisone on day 1 to 5, 29 to 33 and 57 to 61, mercaptopurine for 84 days, Oral MTX on every week and IT MTX on day 1 and 29. Maintenance consists of re-peated 84-day cycles, the total duration of therapy is 2 years for female patients and 3 years for male patients from the start of Interim Maintenance.

2.2.4 CCG1961 protocol

In the induction process, Patients received weekly IV vincristine and IV daunorubicin for 4 weeks, prednisone for 28 days, L-Asparaginase 6 doses every other day from day 9 to 22, IT cytarabine on day 1 and IT MTX on day 8 and 29. In the Consolidation process, Patients received cyclophosphamide on day 1 and 15, IV cytarabine on day 1 to 4, 8 to 11, 15 to 18 and 22 to 25, Oral mercaptopurine for 28ays and IT MTX on day 1, 8, 15 and 22. In the Interim maintenance process, Patients received Oral mercaptopurine for 42days, Oral MTX on day 7, 14, 21, 28 and 35 and IT MTX on day 1, 29. In the Delayed intensification process, Patients received Oral dexamethasone on day 1 to 21, IV vincristine and doxorubicin on day 1, 8 and 15, L-asparaginase 6 doses every other day from day 4, IT MTX on day 1, 29 and 36, cyclophosphamide on day 29, thioguanine on day 29 to 42, IV cytarabine on day 29 to 32 and 36 to 39. In the Maintenance process, Patients received IV vincristine on day 1, 29 and 57, Oral prednisone on day 1 to 5, 29 to 33 and 57 to 61, mercaptopurine for 84 days, Oral MTX weekly for 12 weeks and IT MTX on day 1 and 29. Maintenance consists of repeated 84-day cycles, the total duration of therapy is 2 years for female patients and 3 years for male patients from the start of Interim Maintenance I.

Minor 1: Please describe any ideas the authors have to avoid peripheral neuropathy in the treatment of ALL.

Response 5: Thank you for your valuable comments. In response to your comments, we describe out ideas below.

Glutamic acid is known as an excitatory neurotransmitter that seems to produce a protective activity from VIPN [*] Therefore, the first is taking glutamic acid during treat ALL with vincristine.

In the Second, Vincristine is an essential component of the chemotherapy protocol for treating ALL in children. Therefore, adjusting the dose of vincristine before the onset of peripheral neuropathy can interfere with ALL treatment itself. Therefore, we think it is important to detect it as early as possible and adjust the dose of vincristine in consultation with a pediatrician. We thought of a way to do this. In the simplest way, there is a possibility that the sensory or motor symptoms of peripheral neuropathy may be overlooked by the patient and the caregiver, leading to a late diagnosis. Therefore, it is necessary to accurately educate the patient and caregiver on what sensory or motor symptoms are, inform the attending their doctor immediately if there are even minor symptoms, and conduct follow up nerve conduction study to confirm the presence of peripheral neuropathy. In addition, follow up nerve conduction study is performed immediately before each process (induction, consolidation, etc.). If peripheral neuropathy is confirmed in the nerve conduction study performed at this time, treatment is performed by adjusting the dose of vincristine.

* Triarico, S.;  Romano, A.;  Attina, G.;  Capozza, M. A.;  Maurizi, P.;  Mastrangelo, S.; Ruggiero, A., Vincristine-Induced Peripheral Neuropathy (VIPN) in Pediatric Tumors: Mechanisms, Risk Factors, Strategies of Prevention and Treatment. Int J Mol Sci 2021, 22

Minor 2: Please provide additional information on the usual prognosis for peripheral neuropathy and how it is treated.

Response 6: We appreciate your proper suggestion. In response to your suggestion, we added additional information on the usual prognosis for peripheral neuropathy and how it is treated in “Discussion” section.

(Page8) Discussion, lines 320-239

Chemotherapy-induced peripheral neuropathy (CIPN) is distinct from cancer pain when cancer directly invades nerve tissue and compresses nerves or forms bone metastasis. It is observed in about 40% of patients receiving chemotherapy [17]. In general, if peripheral neuropathy develops after chemotherapy, the symptoms will disappear within a few months after treatment ends. But rarely symptoms may last longer [28]. Treatment for peripheral neuropathy is based on symptoms. If sensory symptoms such as neuropathic pain or paresthesia are the main symptoms, medications such as gabapentin, tricyclic antidepressants, pyridoxine, pyridostigmine, and glutamic acid may be used [29]. In addition, if motor symptoms such as weakness are the main symptom, physical therapy intervention (strength, endurance and balance training) can be performed [30].

[17] Argyriou, A.A.; Bruna, J.; Marmiroli, P.; Cavaletti, G. Chemotherapy-induced peripheral neurotoxicity (CIPN): an update. Crit Rev Oncol Hematol 2012, 82, 51-77

[28] Starobova, H.; Vetter, I. Pathophysiology of Chemotherapy-Induced Peripheral Neuropathy. Front Mol Neurosci 2017, 10, 174

[29] Triarico, S.; Romano, A.; Attina, G.; Capozza, M.A.; Maurizi, P.; Mastrangelo, S.; Ruggiero, A. Vincristine-Induced Peripheral Neuropathy (VIPN) in Pediatric Tumors: Mechanisms, Risk Factors, Strategies of Prevention and Treatment. Int J Mol Sci 2021, 22

[30] Brayall, P.; Donlon, E.; Doyle, L.; Leiby, R.; Violette, K. Physical Therapy-Based

Interventions Improve Balance, Function, Symptoms, and Quality of Life in Patients With

Chemotherapy-Induced Peripheral Neuropathy: A Systematic Review. Rehabil Oncol 2018, 36,

161-166

Thank you again for your detailed opinions about the manuscript. With your sincere advice, this study is able to convey the contents more clearly.

If you think that the above corrections are not appropriate, please let us know.

Thank you.

Ji Yoon Kim, MD, PhD.

Department of Pediatrics

Kyungpook National University Hospital

School of Medicine, Kyungpook National University,

130 Dongdeok-Ro, Jung-Gu, Daegu, 41944, Republic of Korea

Tel: +82-53-200-5704

E-mail: phojyk@knu.ac.kr

Tae-Du Jung, MD, PhD.

Department of Rehabilitation Medicine

Kyungpook National University Chilgok Hospital

School of Medicine, Kyungpook National University

807 Hoguk-ro, Buk-gu, Daegu 41404, Republic of Korea

Tel: +82-53-200-2167

Fax: +82-53-200-2033

Email: teeed0522@hanmail.net

Reviewer 2 Report

The work of Jeong et al. on Vincristine induced Peripheral Neuropathy VINP in children with ALL is an interesting project, however with limited addition to the current knowledge. The research investigates electrophysiologic patterns in clinically symptomatic VIPN.

In addition to the following three major comments, I have some minor comments, see below:

·       This study and what it adds to the current knowledge need to be discussed to a larger extend to the existent work on this topic.

·       This study only included clinically symptomatic VIPN. This needs to be clarified and discussed.

·       The study protocol lacks information (e.g. when, how often electrodiagnostic investigations were performed)

Title

·       I would suggest to modify the title to clarify that this study is based on clinically symptomatic VIPN because all clinically asymptomatic patients who might nevertheless fulfil electrodiagnostic criteria of an axonal VIPN were not investigated

Abstract

·       Is it useful to specify 3 digits after the decimal point?

·       It is confusing to read numbers for the average cumulative dose until VIPN and for the cumulative dose at which in mean (survival curve) a VIPN. I would suggest to focus on the more relevant number in the abstract and describe/discuss different results in more detail in the Results and Discussion section of the main manuscript.

Methods

·       It would be important to know, how many patients who received Vincristine in the same time period were excluded from the study because (as mentioned in the manuscript) they had another condition, e.g. diabetes, that might have an impact on the NCS or because they did not have symptoms indicative for a PNP although treated with Vincristine.

·       Was there a standardized protocol for NCS in all Vincristine treated patients in your center? Because if not, you may have missed VIPN patients who did not show (or mention) “abnormal sensory or motor nerve symptoms after chemotherapy containing Vincristine”. Then: It needs to be mentioned that only patients with defined symptoms for the basis for all calculations (cumulative dosis, survival curves etc.) and that numbers which should include also asymptomatic patients would most likely look different. Little information is given in the results section (should be moved to the Methods section): “All patients underwent an NCS before initiating chemotherapy…”, but there is no information about how often / when additional investigations were performed.

·       “Terminal latency” most likely the more commonly used term is “distal latency”.

·       Was it shown that the normal reference values used as reference for this study reflect 1.) the normal values for the ENMG lab which did the investigations for this study and 2.) the baseline results before Vincristine treatment (stated as being “normal” for all patients within this study before Vincristine treatmet)? Wouldn’t it be more conclusive to use the intraindividual reference (NCS before Vincristine treatment) to decide whether there is an VIPN and to what extend instead of applying an external reference?

Results

·       First sentence: This sentence repeats the inclusion criteria and lacks additional information. I would remove this sentence or add the information when/why the follow-up NCV study was done.

·       In the section “3). Cumulative dose of Vincristine up to diagnosis of VIPN” you should clarify that all calculations are based on clinically symptomatic VIPN because all clinically asymptomatic patients who might nevertheless fulfil electrodiagnostic criteria of an axonal VIPN were not investigated.

·       Kaplan-Meier curve: little information is given about the protocol of NCS. The protocol needs to be described in more detail. If NCS were not done on a routine basis after Vincristine treatment but only when symptoms of a VINP appeared, then this Kaplan-Meier curve rather describes the appearance of symptoms of a VINP and not the VIPN itself which most likely would have been detect far earlier and with a lower cumulative Vincristine dose by electrodiagnostic investigations.

Discussion

·       The results of this study need to be discussed in the larger context of similar studies which are plenty. Just to mention a few: https://doi.org/10.3389/fphar.2021.771487, https://doi.org/10.1016/j.critrevonc.2017.04.004, https://doi.org/10.1212/01.WNL.0000154642.45474.28, https://doi.org/10.1177/0883073813491829, https://doi.org/10.17724/jicna.2017.75

·       Again, it needs to be discussed, that this study is only about clinically symptomatic VIPN.

·       The discussion about the maturation of the peripheral nerve system as a major reason for the pronounced motor damage in childhood VINP is quite vague and lacks referenced information. The same applies the discussion of potential similarities to the axonal variant of GBS.

·       I do not understand the following sentence: “Nevertheless, this study shows that VIPN can be diagnosed early before neurological symptoms occur…”, because no data are described in this study related to clinically asymptomatic VIPN and only clinically symptomatic patients were included.

Author Response

Response to Reviewer 2 Comments

Dear Reviewer #2,

Thank you very much for your kind letter and comments concerning our manuscript entitled Electrophysiologic patterns of Vincristine induced Peripheral Neuropathy in Children with ALL”. First of all, thank you very much for valuing the strengths of our article.

The comments were valuable and very helpful in critically revising and improving our paper. We have discussed the comments carefully and have made corresponding corrections which we hope will be met with approval. The following are responses to your comments:

Point 1: This study and what it adds to the current knowledge need to be discussed to a larger extend to the existent work on this topic.

Response 1: Thank you for your valuable comment. We tried to discuss to a larger extend to the existent work on this topic in “Discussion” section

(Page 8) Discussion, lines 305-319

Previous studies have reported that the characteristics are more similar to axonal type of neuropathy than to demyelinating type of neuropathy [8,24], which was confirmed in the NCS of children with VIPN in this study Given that the electrophysiological characteristics are similar to acute motor axonal neuropathy, a variant of Guillain–Barre syndrome, the effect of vincristine on nerves in children may be similarly related to the mechanism of channelopathy.

Additionally, this study analyzed the cumulative incidence of VIPN according to the cumulative dose of vincristine using the survival curve. According to the analysis, peripheral neuropathy occurred in 50% of the patients when 15.5 ± 1.77 mg/m2 of vincristine was administered cumulatively. Several previous studies have also found that the incidence of VIPN increases in proportion to the dose of vincristine used in children diagnosed with VIPN [25-27]. However, this study presented a specific value of the VIPN in-cidence according to the cumulative dose of vincristine for which the survival curve was applied. Hence, the objective nature of the analysis might be considered an advantage in the present study compared to the previous studies

[8] Jain, P.; Gulati, S.; Seth, R.; Bakhshi, S.; Toteja, G.S.; Pandey, R.M. Vincristine-induced neuropathy in childhood ALL (acute lymphoblastic leukemia) survivors: prevalence and electrophysiological characteristics. Journal of child neurology 2014, 29, 932-937

[24] Reinders-Messelink, H.A.; Van Weerden, T.W.; Fock, J.M.; Gidding, C.E.; Vingerhoets, H.M.; Schoemaker, M.M.; Goeken, L.N.; Bokkerink, J.P.; Kamps, W.A. Mild axonal neuropathy of children during treatment for acute lymphoblastic leukaemia. European journal of paediatric neurology : EJPN : official journal of the European Paediatric Neurology Society 2000, 4, 225-233

[25] Lavoie Smith, E.M.; Li, L.; Hutchinson, R.J.; Ho, R.; Burnette, W.B.; Wells, E.; Bridges, C.; Renbarger, J. Measuring vincristine-induced peripheral neuropathy in children with acute lymphoblastic leukemia. Cancer nursing 2013, 36, E49-60

[26] Diouf, B.; Crews, K.R.; Lew, G.; Pei, D.; Cheng, C.; Bao, J.; Zheng, J.J.; Yang, W.; Fan, Y.; Wheeler, H.E.; et al. Association of an inherited genetic variant with vincristine-related peripheral neuropathy in children with acute lymphoblastic leukemia. Jama 2015, 313, 815-823

[27] Guilhaumou, R.; Solas, C.; Bourgarel-Rey, V.; Quaranta, S.; Rome, A.; Simon, N.; Lacarelle, B.; Andre, N. Impact of plasma and intracellular exposure and CYP3A4, CYP3A5, and ABCB1 genetic polymorphisms on vincristine-induced neurotoxicity. Cancer chemotherapy and pharmacology 2011, 68, 1633-1638

Point 2: This study only included clinically symptomatic VIPN. This needs to be clarified and discussed.

Response 2: Thank you for your valuable comments. The department of rehabilitation medicine at our center performed nerve conduction tests on children with ALL who were consulted from the pediatric department. The time of consultation was when ALL was first diagnosed and symptoms of peripheral neuropathy were present. Therefore, it was inevitable to conduct a retrospective study rather than a study, and we dealt with this limitation in the discussion section. In response to your comment, we clarified that only symptomatic patients were included in the “Materials and Methods” section.

(Page 2) Materials and Methods, lines 69-75

This study included patients who had visited Chilgok Kyungpook National Univer-sity Hospital between January 2012 and December 2021, were diagnosed with ALL, and received chemotherapy. The department of Rehabilitation Medicine at our center performed NCS on children with ALL who were consulted from the pediatric department when patients occurred symptoms of peripheral neuropathy after chemotherapy containing vincristine. Therefore, we inevitably retrospectively investigated the information of only symptomatic patients with abnormal findings in the NCS.

And, we discussed our limitation that only symptomatic patients were included in “Discussion” section

(Page 9) Discussion, lines 360-368

Second, this study retrospectively analyzed NCS data from children who showed symptoms of peripheral neuropathy while receiving chemotherapy. Asymptomatic VIPN patients were not included in our study. If they were included, the results of our study might have changed. Therefore, for an accurate analysis of the VIPN incidence according to the cumulative dose, analysis of data from an NCS performed regularly, regardless of neurological symptoms, is required. Therefore, it is necessary to set a certain cumulative dose or time interval and conduct the test at an appropriate time. If a prospective study that supplements these limitations is undertaken in the future, more meaningful results may be obtained.

Point 3: The study protocol lacks information (e.g. when, how often electrodiagnostic investigations were performed)

Response 3: Thank you for your valuable comments. The department of Rehabilitation Medicine at our center performed NCS on children with ALL who were consulted from the pediatric department when patients occurred symptoms of peripheral neuropathy after chemotherapy containing vincristine. Therefore, our center did not have NCS protocol for ALL patients who received chemotherapy.

Through this study, we could know early diagnosis of VIPN is very important and the necessity of the NCS protocol for ALL patients who received chemotherapy was discussed in the “Discussion” and “Conclusions” section

(Page 9) Discussion, lines 363-367

Therefore, for an accurate analysis of the VIPN incidence according to the cumulative dose, analysis of data from an NCS performed regularly, regardless of neurological symptoms, is required. Therefore, it is necessary to set a certain cumulative dose or time interval and conduct the test at an appropriate time

(Page 9) Conclusions, lines 376-379

VIPN causes a delay in treatment and lowers the quality of life of children. Therefore, if VIPN is diagnosed in the early period of chemotherapy through regular NCS and appro-priate management is performed, the quality of treatment can be improved.

<Title>

Point 4: I would suggest to modify the title to clarify that this study is based on clinically symptomatic VIPN because all clinically asymptomatic patients who might nevertheless fulfil electrodiagnostic criteria of an axonal VIPN were not investigated

Response 4: We appreciate your proper suggestion. In response to your suggestion, We changed the title like below.

(Page1) Title, lines 2-4

Electrophysiologic patterns of Vincristine-induced Peripheral Neuropathy in children with ALL

→ Electrophysiologic patterns of Symptomatic Vincristine-induced Peripheral Neuropathy in Children with Acute Lymphocytic Leukemia.

<Abstract>

Point 5: Is it useful to specify 3 digits after the decimal point? It is confusing to read numbers for the average cumulative dose until VIPN and for the cumulative dose at which in mean (survival curve) a VIPN. I would suggest to focus on the more relevant number in the abstract and describe/discuss different results in more detail in the Results and Discussion section of the main manuscript.

Response 5: We appreciate your proper suggestion. First, we are sorry for confusing to read article. Several previous studies have also found that the incidence of VIPN increases in proportion to the dose of vincristine used in children diagnosed with VIPN[22-24]. However, this study presented a specific value of the VIPN incidence according to the cumulative dose of vincristine for which the survival curve was applied. In the survival curve, we considered the average cumulative dose and the median value, the cumulative amount of vincristine at which approximately 50% of patients developed VIPN, to be the most significant. Therefore, we mentioned the average cumulative dose and the median value in the survival curve in the abstract. As it’s confusing to read, we changed it to specify 2 digits after the decimal point.

(Page 1) Abstract, lines 25-26

The average cumulative dose until diagnosis of vincristine-induced peripheral neuropathy was 14.993 ± 1.209 mg/m2

→ The average cumulative dose until diagnosis of vincristine-induced peripheral neuropathy was 14.99 ± 1.21 mg/m2

(Page 1) Abstract, line 29-30

about 50% of children developed peripheral neuropathy at the dose of 15.497±1.773 mg/m2

→ about 50% of children developed peripheral neuropathy at the dose of 15.5±1.77 mg/m2

 <Method>

Point 6: It would be important to know, how many patients who received Vincristine in the same time period were excluded from the study because (as mentioned in the manuscript) they had another condition, e.g. diabetes, that might have an impact on the NCS or because they did not have symptoms indicative for a PNP although treated with Vincristine.

Response 6: Thank you for your valuable comment. In the same time period, we performed nerve conduction study only for symptomatic patients referred from pediatrics. Among them, we excluded patients with another condition, e.g. diabetes, that might have an impact on the NCS. Unfortunately, it is not known how many children received vincristine during the same period.

Point 7: Was there a standardized protocol for NCS in all Vincristine treated patients in your center? Because if not, you may have missed VIPN patients who did not show (or mention) “abnormal sensory or motor nerve symptoms after chemotherapy containing Vincristine”. Then: It needs to be mentioned that only patients with defined symptoms for the basis for all calculations (cumulative dosis, survival curves etc.) and that numbers which should include also asymptomatic patients would most likely look different.

Response 7: We appreciate your proper suggestion. Unfortunately, prior to this study, there was no standardized protocol for NCS in all Vincristine treated patients in out center. After this study, we felt the need for a standardized protocol to early diagnose or prevent in Vincristine treated patients, and we plan to consult with pediatricians. I agree with your comment that because only the NCS results and cumulative dose of vincristine in symptomatic patients were enrolled in the statistics, the test results may differ if asymptomatic VIPN patients are also included. We clarify this comments in “Discussion” section.

(Page9) Discussion, lines 362-363

Asymptomatic VIPN patients were not included in our study. If they were included, the results of our study might have changed.

Point 8: Little information is given in the results section (should be moved to the Methods section): “All patients underwent an NCS before initiating chemotherapy…”, but there is no information about how often / when additional investigations were performed.

Response 8: Thank you for your valuable comment. To clarify the answer to this comment and the above comments in methods section(point 6 and 7) , we moved some sentences from the “Result” section and revised overall “Subjects” subsection.

(Page 2) Materials and Methods, lines 68-82

2.1. Subjects

This study included patients who had visited Chilgok Kyungpook National Univer-sity Hospital between January 2012 and December 2021, were diagnosed with ALL, and received chemotherapy. The department of Rehabilitation Medicine at our center performed NCS on children with ALL who were consulted from the pediatric department when patients occurred symptoms of peripheral neuropathy after chemotherapy containing vincristine. Therefore, we inevitably retrospectively investigated the information of only symptomatic patients with abnormal findings in the NCS. The data included the chemotherapy protocol used, cumulative dose of vincristine until the NCS, and results of the NCS. In total, 30 patients were analyzed based on the inclusion criteria. The inclusion criteria were as follows: (1) patients who were younger than 18 years of age; (2) Patients diagnosed with only ALL and not other types of leukemia; (3) patients had one or more nerve abnormalities in the NCS. The exclusion criteria were as follows; (1) patients had peripheral polyneuropathy before chemotherapy; (2) patients had underlying diseases, that can influence NCS results, such as diabetes; (3) patients had unstable vital sign.   

Point 9: “Terminal latency” most likely the more commonly used term is “distal latency”.

Response 9: We appreciate your proper suggestion. In response to your suggestion, we revised terminal latency to distal latency.

(Page 4) Materials and Methods, lines 174-176

In motor nerves, the amplitude and distal latency of compound muscle action potential (CMAP) and conduction velocity between stimulation points were measured.

(Page 4) Materials and Methods, lines 178-180

Distal latency was measured in milliseconds as the time from stimulation of the motor nerve to the onset of CMAP, and conduction velocity was measured in meters per second.

Point 10:  Was it shown that the normal reference values used as reference for this study reflect 1.) the normal values for the EMG lab which did the investigations for this study and 2.) the baseline results before Vincristine treatment (stated as being “normal” for all patients within this study before Vincristine treatmet)? Wouldn’t it be more conclusive to use the intraindividual reference (NCS before Vincristine treatment) to decide whether there is an VIPN and to what extend instead of applying an external reference?

Response 10: Thank you for your valuable comment. Until now, several discussions have been made about the definition of peripheral polyneuropathy. Normally, when the CMAP amplitude is lower than the normal reference, when the distal latency is longer than the normal reference, and when the conduction velocity is lower than the normal reference, it can be considered that there is an injury to the peripheral nerve. In our center, children with injuries in more than one peripheral nerve were included as mentioned at method section. The criterion for having an injury in the peripheral nerve was set based on the normal reference.

We are impressed with your idea of using intraindividual references instead of normal reference and we think it is very ingenious. So, if the opportunity arises in the future, we will conduct research using intraindividual references as well.

<Results>

Point 11: First sentence: This sentence repeats the inclusion criteria and lacks additional information. I would remove this sentence or add the information when/why the follow-up NCV study was done.

Response 11: We appreciate your proper suggestion. In response to your suggestion, we moved some sentences from the “Result” section to the “Materials and Methods” section and revised overall “Subjects” subsection.

(Page 2) Materials and Methods, lines 68-82

2.1. Subjects

This study included patients who had visited Chilgok Kyungpook National Univer-sity Hospital between January 2012 and December 2021, were diagnosed with ALL, and received chemotherapy. The department of Rehabilitation Medicine at our center performed NCS on children with ALL who were consulted from the pediatric department when patients occurred symptoms of peripheral neuropathy after chemotherapy containing vincristine. Therefore, we inevitably retrospectively investigated the information of only symptomatic patients with abnormal findings in the NCS. The data included the chemotherapy protocol used, cumulative dose of vincristine until the NCS, and results of the NCS. In total, 30 patients were analyzed based on the inclusion criteria. The inclusion criteria were as follows: (1) patients who were younger than 18 years of age; (2) Patients diagnosed with only ALL and not other types of leukemia; (3) patients had one or more nerve abnormalities in the NCS. The exclusion criteria were as follows; (1) patients had peripheral polyneuropathy before chemotherapy; (2) patients had underlying diseases, that can influence NCS results, such as diabetes; (3) patients had unstable vital sign.   

(Page 5) Results, lines 202-209

In total, 30 patients were analyzed based on the inclusion criteria. Among children with different types of leukemia, only those diagnosed with ALL were included. All patients underwent an NCS before initiating chemotherapy, and no abnormal findings were found. The demographic data of the patients are presented in Table 1. There were 15 boys and 15 girls, and the average age was 7.63 ± 4.69 years. The average cumulative dose of vincristine until abnormal findings were detected in the NCS was 14.99 ± 1.21 mg/m2. The average period from the first administration of vincristine to the discovery of abnor-mal findings in the NCS was 143.37 ± 74.02 days.

→ The demographic data of the patients are presented in Table 1. There were 15 boys and 15 girls, and the average age was 7.63 ± 4.69 years. The average cumulative dose of vincristine until abnormal findings were detected in the NCS was 14.99 ± 1.21 mg/m2. The average period from the first administration of vincristine to the discovery of abnor-mal findings in the NCS was 143.37 ± 74.02 days.

Point 12: In the section “3). Cumulative dose of Vincristine up to diagnosis of VIPN” you should clarify that all calculations are based on clinically symptomatic VIPN because all clinically asymptomatic patients who might nevertheless fulfil electrodiagnostic criteria of an axonal VIPN were not investigated.

Response 12: We appreciate your proper suggestion. In response to your comment, we changed the title of this subsection to carify that all calculations are based on clinically symptomatic VIPN.

(Page 6) Results, line 253

3.3. Cumulative dose of vincristine up to diagnosis of VIPN 3.3. Cumulative dose of vincristine up to diagnosis of symptomatic VIPN

Point 13: Kaplan-Meier curve: little information is given about the protocol of NCS. The protocol needs to be described in more detail. If NCS were not done on a routine basis after Vincristine treatment but only when symptoms of a VINP appeared, then this Kaplan-Meier curve rather describes the appearance of symptoms of a VINP and not the VIPN itself which most likely would have been detect far earlier and with a lower cumulative Vincristine dose by electrodiagnostic investigations.

Response 13: We appreciate your proper suggestion. Unfortunately, there was no standardized protocol for NCS in all Vincristine treated patients, so we couldn’t describe the protocol in detail. As you suggested, We changed it to make our Kaplan-Meire curve describe rather the appearance of symptomatic VIPN and not the VIPN itself more clear.

(Page 6) Results, lines 255-257

According to the aforementioned statistical analysis, symptomatic VIPN occurred in approximately 50% of the patients when the cumulative dose of vincristine administered was 15.5 ± 1.77 mg/m2.

(Page 6) Results, lines 257-260

When analyzed using the survival table, symptomatic VIPN occurred in approximately 25% and 75% of the patients at a cumulative vincristine dose of 10.48 ± 2.42 and 19.21 ± 2.95 mg/m2, respectively.

(Page 7) Results, lines 262-264

Kaplan–Meier curves for the relationship between the cumulative dose of vincristine and occurrence of symptomatic VIPN. The median dose of vincristine until onset symptoms of VIPN is 15.5 ± 1.77 mg/m2.

<Discussion>

Point 14: The results of this study need to be discussed in the larger context of similar studies which are plenty. Just to mention a few:

https://doi.org/10.3389/fphar.2021.771487,https://doi.org/10.1016/j.critrevonc.2017.04.00https://doi.org/10.1212/01.WNL.0000154642.45474.28,https://doi.org/10.1177/0883073813491829, https://doi.org/10.17724/jicna.2017.75

Response 14: Thank you for your valuable comment. In the similar context as point 1, we tried to discuss to a larger extend to the existent work on this topic in “Discussion” section

(Page 8) Discussion, lines 305-319

Previous studies have reported that the characteristics are more similar to axonal type of neuropathy than to demyelinating type of neuropathy [8,24], which was confirmed in the NCS of children with VIPN in this study Given that the electrophysiological characteristics are similar to acute motor axonal neuropathy, a variant of Guillain–Barre syndrome, the effect of vincristine on nerves in children may be similarly related to the mechanism of channelopathy.

Additionally, this study analyzed the cumulative incidence of VIPN according to the cumulative dose of vincristine using the survival curve. According to the analysis, peripheral neuropathy occurred in 50% of the patients when 15.5 ± 1.77 mg/m2 of vincristine was administered cumulatively. Several previous studies have also found that the incidence of VIPN increases in proportion to the dose of vincristine used in children diagnosed with VIPN [25-27]. However, this study presented a specific value of the VIPN in-cidence according to the cumulative dose of vincristine for which the survival curve was applied. Hence, the objective nature of the analysis might be considered an advantage in the present study compared to the previous studies

[8] Jain, P.; Gulati, S.; Seth, R.; Bakhshi, S.; Toteja, G.S.; Pandey, R.M. Vincristine-induced neuropathy in childhood ALL (acute lymphoblastic leukemia) survivors: prevalence and electrophysiological characteristics. Journal of child neurology 2014, 29, 932-937

[24] Reinders-Messelink, H.A.; Van Weerden, T.W.; Fock, J.M.; Gidding, C.E.; Vingerhoets, H.M.; Schoemaker, M.M.; Goeken, L.N.; Bokkerink, J.P.; Kamps, W.A. Mild axonal neuropathy of children during treatment for acute lymphoblastic leukaemia. European journal of paediatric neurology : EJPN : official journal of the European Paediatric Neurology Society 2000, 4, 225-233

[25] Lavoie Smith, E.M.; Li, L.; Hutchinson, R.J.; Ho, R.; Burnette, W.B.; Wells, E.; Bridges, C.; Renbarger, J. Measuring vincristine-induced peripheral neuropathy in children with acute lymphoblastic leukemia. Cancer nursing 2013, 36, E49-60

[26] Diouf, B.; Crews, K.R.; Lew, G.; Pei, D.; Cheng, C.; Bao, J.; Zheng, J.J.; Yang, W.; Fan, Y.; Wheeler, H.E.; et al. Association of an inherited genetic variant with vincristine-related peripheral neuropathy in children with acute lymphoblastic leukemia. Jama 2015, 313, 815-823

[27] Guilhaumou, R.; Solas, C.; Bourgarel-Rey, V.; Quaranta, S.; Rome, A.; Simon, N.; Lacarelle, B.; Andre, N. Impact of plasma and intracellular exposure and CYP3A4, CYP3A5, and ABCB1 genetic polymorphisms on vincristine-induced neurotoxicity. Cancer chemotherapy and pharmacology 2011, 68, 1633-1638

Point 15: Again, it needs to be discussed, that this study is only about clinically symptomatic VIPN.

Response 15: Thank you for your valuable comment. In the similar context as point 2, the department of rehabilitation medicine at our center performed nerve conduction tests on children with ALL who were consulted from the pediatric department. The time of consultation was when ALL was first diagnosed and symptoms of peripheral neuropathy were present. Therefore, it was inevitable to conduct a retrospective study rather than a study, and we dealt with this limitation in the discussion section. In response to your comment, we clarified that only symptomatic patients were included in the “Materials and Methods” section.

(Page 2) Materials and Methods, lines 69-75

This study included patients who had visited Chilgok Kyungpook National Univer-sity Hospital between January 2012 and December 2021, were diagnosed with ALL, and received chemotherapy. The department of Rehabilitation Medicine at our center performed NCS on children with ALL who were consulted from the pediatric department when patients occurred symptoms of peripheral neuropathy after chemotherapy containing vincristine. Therefore, we inevitably retrospectively investigated the information of only symptomatic patients with abnormal findings in the NCS.

And, we discussed our limitation that only symptomatic patients were included in “Discussion” section

(Page 9) Discussion, lines 360-368

Second, this study retrospectively analyzed NCS data from children who showed symptoms of peripheral neuropathy while receiving chemotherapy. Asymptomatic VIPN patients were not included in our study. If they were included, the results of our study might have changed. Therefore, for an accurate analysis of the VIPN incidence according to the cumulative dose, analysis of data from an NCS performed regularly, regardless of neurological symptoms, is required. Therefore, it is necessary to set a certain cumulative dose or time interval and conduct the test at an appropriate time. If a prospective study that supplements these limitations is undertaken in the future, more meaningful results may be obtained.

Point 16: The discussion about the maturation of the peripheral nerve system as a major reason for the pronounced motor damage in childhood VINP is quite vague and lacks referenced information. The same applies the discussion of potential similarities to the axonal variant of GBS.

Response 16: Thank you for your valuable comment. In response to your comment, additional discussions and references on major reason for the pronounced motor damage in childhood VIPN were added to the “Discussion” section.

(Page 7,8) Discussion, lines 288-296

In general, peripheral neuropathy occurring after vincristine administration mainly involves sensory nerves [17,18], but in this study, motor nerves were more often involved. This is a characteristic of pediatric VIPN that differs from characteristics of adult VIPN, and we can consider the temporal difference in myelination in the maturation process of the central and peripheral nerves in the developmental process[19]. It is known that the maturation and development of sensory nerves precede motor nerves[20,21]. Therefore, it is presumed that motor nerves, which are in a relatively immature myelination process, are susceptible to damage by drugs such as vincristine that cause axon damage, which explains the remarkable findings of characteristic motor nerve invasion in children[19]

[17] Argyriou, A.A.; Bruna, J.; Marmiroli, P.; Cavaletti, G. Chemotherapy-induced peripheral neurotoxicity (CIPN): an update. Crit Rev Oncol Hematol 2012, 82, 51-77

[18] Holland, J.F.; Scharlau, C.; Gailani, S.; Krant, M.J.; Olson, K.B.; Horton, J.; Shnider, B.I.; Lynch, J.J.; Owens, A.; Carbone, P.P.; et al. Vincristine treatment of advanced cancer: a cooperative study of 392 cases. Cancer research 1973, 33, 1258-1264.

[19] Courtemanche, H.; Magot, A.; Ollivier, Y.; Rialland, F.; Leclair-Visonneau, L.; Fayet, G.; Camdessanche, J.P.; Pereon, Y. Vincristine-Induced Neuropathy: Atypical Electrophysiological Patterns in Children. Muscle Nerve 2015, 52, 981-985

[20] Ino, D.; Iino, M. Schwann cell mitochondria as key regulators in the development and

maintenance of peripheral nerve axons. Cell Mol Life Sci 2017, 74, 827-835

[21] Ibanez, C.F.; Ernfors, P. Hierarchical control of sensory neuron development by neurotrophic factors. Neuron 2007, 54, 673-675

Point 17: I do not understand the following sentence: “Nevertheless, this study shows that VIPN can be diagnosed early before neurological symptoms occur…”, because no data are described in this study related to clinically asymptomatic VIPN and only clinically symptomatic patients were included.

Response 17: We appreciate your proper suggestion. Because we only included patients with symptomatic VIPN, I agree with you that you don’t understand the sentence you said. Through a retrospective study using the survival curve, what we thought was a future task is that based on these data, it is more meaningful to closely monitor the patient’s condition when vincristine is administered in the cumulative dose range in which the risk of VIPN is increased and to suggest an appropriate period to conduct a NCS. We want to convey that NCS can diagnose VIPN before neurologic symptoms develop. Therefore, we changed the sentence you pointed out to suggest the reason for it.

(Page 8) Discussion, lines 333-338

Nevertheless, this study shows that VIPN can be diagnosed early before neurological symptoms occur

→ In children, since movement disorders and involvement of motor nerves appear relatively early and severely, NCS can be useful. Because NCS preferentially represent motor nerve lesion, which are large nerve fibers.

Thank you again for your detailed opinions about the manuscript. With your sincere advice, this study is able to convey the contents more clearly.

If you think that the above corrections are not appropriate, please let us know.

Thank you.

Ji Yoon Kim, MD, PhD.

Department of Pediatrics

Kyungpook National University Hospital

School of Medicine, Kyungpook National University,

130 Dongdeok-Ro, Jung-Gu, Daegu, 41944, Republic of Korea

Tel: +82-53-200-5704

E-mail: phojyk@knu.ac.kr

Tae-Du Jung, MD, PhD.

Department of Rehabilitation Medicine

Kyungpook National University Chilgok Hospital

School of Medicine, Kyungpook National University

807 Hoguk-ro, Buk-gu, Daegu 41404, Republic of Korea

Tel: +82-53-200-2167

Fax: +82-53-200-2033

Email: teeed0522@hanmail.net

Round 2

Reviewer 1 Report

I received sincere replies to several detailed questions and answers from the author. The limitation has been added, so the paper will be scientifically changed credible and reliable.

The reviewers suggest a final addition to one of the following points.

Many drugs can cause peripheral neuropathy, including vinorelbine, vinblastine, carboplatin, cisplatin, paclitaxel, docetaxel and cabazitaxel. However, the protocols for AALL0331, AALL0232, AALL0434 and CCG1961, which were performed in the treatment of ALL in this study, did not use drugs other than these vincristines, so the effects of drugs other than vincristine were not considered.

Please add these points to the discussion as well.

Best regards

Dr. Reviewer

Reviewer 2 Report

the authors have responded adequately to my comments and modified the manuscript accordingly. I do not have further comments.